# Continuous Wireless Vital Sign Sensors for Detecting Severe Deviations at a Transitional Care Facility—An Observational Feasibility Study

**DOI:** 10.3390/s25226970

**Published:** 2025-11-14

**Authors:** Jesper Mølgaard, Camilla Haahr-Raunkjaer, Søren Straarup Rasmussen, Loraine Villacorte Andersen, Christian S. Meyhoff, Eske K. Aasvang

**Affiliations:** 1Department of Anesthesiology, Center for Cancer and Organ Diseases, Copenhagen University Hospital, Rigshospitalet, 2100 Copenhagen, Denmark; 2Biomedical Signal Processing and AI Group, Digital Health Section, Department of Health Technology, Technical University of Denmark, 2800 Kgs Lyngby, Denmark; 3Bystævneparken, Center for Rehabilitation and Acute Care, 2700 Brønshøj, Denmark; 4Department of Anaesthesia and Intensive Care, Copenhagen University Hospital -Bispebjerg and Frederiksberg, 2200 Copenhagen, Denmark; 5Department of Clinical Medicine, University of Copenhagen, 2100 Copenhagen, Denmark

**Keywords:** transitional care, remote monitoring, vital sign values, readmission

## Abstract

After hospital admissions, patients may be discharged to a transitional care facility if they are considered physically or mentally unable to be at home. This study aimed to assess the feasibility of continuous wearable vital sign monitoring (CVSM) at a low-staffed transitional care facility. This was a feasibility study exploring CVSM (blinded for staff) for up to 96 h in 20 patients discharged to a transitional care facility. Feasibility was assessed by monitoring duration. Secondarily, deviating vital sign values from both CVSM and manual vital sign monitoring were described. Twenty patients were continuously monitored for 1565 h in total. The median duration of monitoring with at least one device was 80 h (IQR 50–93), equivalent to 83% of the monitored period. The median cumulative duration of abnormal vital signs was 620 min [IQR 336 min–930 min]. The transitional care facility staff did not assess vital signs in any patients during the monitored period. Forty percent of patients developed at least one medical complication and were readmitted to the hospital.

## 1. Introduction

In 2015, an analysis of 6.8 million hospital admissions found that the risk of hospital readmissions within 30 days was between 13 and 18% [1]. This is consistent with previous research demonstrating a readmission risk up to 16–22% for patients discharged within 30 days of acute or subacute surgery [2,3,4]. A systematic review using a Monte Carlo simulation estimated the mean price of a readmission to be USD 16,037. That results in an annual cost of approximately USD 15 billion for readmissions alone [5].

The length of hospital stay has decreased substantially over the last decades, in part due to a focus on enhanced recovery after surgery [6,7,8] and similar approaches across medical specialties. However, this has resulted in a situation where the only patients remaining in hospitals for prolonged periods are those requiring advanced medical treatment and close observation [9,10]. A potentially frail subgroup of patients is discharged to transitional care facilities to recover, rehabilitate, and regain performance if discharge to their own home is deemed unsafe although no additional hospitalization is required.

In comparison with hospital wards, where vital signs are monitored regularly every 6–12 h and interventions are available in the event of patient deterioration [11], transitional care facilities monitor vital signs only when deemed pertinent. As a result, patient deterioration may go unrecognized, resulting in the progression of complications associated with otherwise manageable conditions, ultimately resulting in irreversible injury and/or hospital readmission. The feasibility and capability of detecting otherwise unnoticed abnormal vital signs in hospitalized patients have been demonstrated [12,13,14,15,16,17], including a significant impact in reducing adverse events [18,19]. However, the feasibility and potential impact of wearable devices for 24/7 continuous vital sign monitoring (CVSM) in patients admitted to transitional care facilities with lower staffing than hospital wards have not been explored. Therefore, we aimed to investigate the feasibility of continuous wearable devices and the frequency of abnormal vital signs in patients admitted to a transitional care facility. We hypothesized that CVSM with at least one device would be acceptable for the patients for at least two thirds of the scheduled time.

## 2. Methods

### 2.1. Study Design and Participants

Prior to patient inclusion, approvals from the Regional Ethics Committee (approval number H-19031634) and the Data Protection Agency were obtained. The study was registered at https://clinicaltrials.gov (accessed on 4 November 2025) as NCT05345626 and conducted in accordance with the STROBE checklist. Patients were recruited and included after obtaining written informed consent from October 2019 to September 2021 with the exception of the period from March 2020 to March 2021 due to the COVID-19 pandemic.

The inclusion criteria were admission to the transitional care facility following hospitalization for surgery or ischemic heart disease, chronic heart disease, stroke, chronic obstructive pulmonary disease, or diabetes mellitus. Patients were enrolled within 72 h after discharge from the hospital and transfer to the transitional care facility. Patients were excluded if they could not provide informed consent, had an implanted pacemaker or cardioverter–defibrillator, or were allergic to plaster, plastic, or silicone.

Following inclusion, the wearable monitoring system was immediately attached to the patient, and specific vital signs were recorded.

### 2.2. Resources and Daily Routine at the Transitional Care Facility

The transitional care facility had a total capacity of 70 beds. Each patient was assigned their own room and equipped with a manual alarm to use if assistance was required. Personal hygiene assistance was provided as needed, and physiotherapists exercised with patients several times a week. Two nurses were in charge of medication administration, clinical assessments requested by care workers, and communication with external physicians if deemed necessary. The average length of stay at the specific transitional care facility was 24 days.

### 2.3. Description of the Wireless Devices and Abnormal Vital Sign Thresholds

Three CE and FDA certified and clinically validated [20,21] wireless and wearable devices continuously recorded five different vital signs (Figure 1).

A. Isansys Lifetouch, measuring heart rate (HR) and respiratory rate (RR); B. Meditech BlueBP-05, measuring blood pressure (BP); C. Nonin, measuring peripheral oxygen saturation (SpO_2_) and pulse rate (PR).

Heart rate and respiratory rate was recorded using Isansys Lifetouch (Isansys Lifecare, Oxfordshire, UK), a single-lead ECG (electrocardiogram) patch, recording 1000 samples per second. It measures the heart rate (HR/min) sampled per minute by R-peak interval detection. The respiration rate (RR/min) is derived from changes in the amplitude of the QRS complex due to changes in thoracic impedance. Blood pressure was recorded using the Meditech BlueBP-05 (Meditech Ltd., Budapest, Hungary) is an upper arm cuff-based programable oscillometric blood pressure (BP) device. It was programmed to take measurements every 30 min throughout the day (07:00–21:59) and every 60 min at night (22:00–06:59). Lastly, peripheral blood oxygenation was recorded using the Nonin WristOx2 3150 (Nonin Medical inc., Minnesota, USA) is a finger pulse oximeter comprising a soft finger sensor (8000SM-WO2) connected to a wrist-borne device via a cable. It measures the SpO_2_ and the PR at 1 Hz from the four previously detected beats. Values were averaged per minute. Data were transmitted via Bluetooth to a patient gateway (Isansys Lifecare, Oxfordshire, UK) positioned next to the patient, immediately storing HR, RR, PR, and SpO_2_ (BP was transferred manually every day). The ECG patch kept recorded values when out of Bluetooth reach for later automatic transfer, which was not the case for the SpO_2_ device. Care workers were blinded to the data. We encouraged participants and care workers not to remove the equipment, except for personal hygiene or when attending a physiotherapy session, making wearing the SpO_2_ and the BP device challenging. Once daily, study personnel visited participants to ensure data quality, change batteries, and encourage patient compliance.

### 2.4. Abnormal Vital Signs

Thresholds defining abnormal vital signs were inspired by and modeled on the Early Warning Scoring (EWS) systems [11]. To reduce irrelevant alarms, we combined the thresholds with subjectively selected durations according to the severity of the vital sign abnormality, a two-dimensional approach termed “sustained episodes of abnormal vital signs.” For example, RR > 24 min^−1^ had to be sustained for ≥5 min to qualify as a sustained episode, whereas RR > 30 min^−1^ had to be sustained for ≥1 min. Furthermore, if two episodes occurred within 5 min, they were registered as one episode despite a period of non-abnormal values in between.

### 2.5. Outcomes

The primary outcome was the fraction of the monitoring duration with up-time for at least one device divided by the planned monitoring duration. Up-time only included periods with data transmission of valid data. We did not include duration with intermittent data loss. The secondary outcomes were as follows: (1) the duration of predefined abnormal vital signs measured by wearable devices and by the transitional care facility manually; (2) the percentage of patients with any predefined episodes of abnormal vital signs; (3) the percentage of patients developing a complication (defined as a physiologic decline requiring intervention from a doctor) following 30 days of study inclusion; (4) the percentage of patients readmitted to the hospital 30 days after study inclusion. Lastly, we chose one patient with a complication to illustrate the vital signs during the monitoring period.

### 2.6. Statistical Analysis

Due to the exploratory and hypothesis-generating nature of this feasibility study, not intended to test clinical effectiveness, we did not perform a formal power calculation for the effect size. Instead we selected a pragmatic sample size of 20 participants to obtain preliminary estimates of feasibility outcomes (recruitment and retention rates, device adherence, data completeness, and safety signals) within our available recruitment period and resources. Data are presented as percentages and numbers or as medians and interquartile ranges (IQRs). Descriptive statistics were used to assess patient demographics and the feasibility of monitoring in a transitional care facility, and hospital and transitional care facility data were gathered from electronic medical records. The REDCap (Vanderbilt University, Nashville, TN, USA) data collection tool was used to manage data, and python v. 3.13.7 (Python Software Foundation, Wilmington, DE, USA) with pandas v 2.3.2 (the pandas development team) was used for data analysis.

## 3. Results

### 3.1. Patient Population

We screened 61 potentially eligible patients, of which 21 patients refused to participate, per the ethical guidelines, and 20 patients were excluded by the investigator. The remaining 20 patients were found to be eligible and consented to participating (Figure 2).

Patients declining to participate were the main reason for exclusion. The participants were primarily females (60%), the median age was 83 years [IQR 74–89], and 75% had an American Society of anesthesiologists Classification (ASA) score of III (Table 1).

### 3.2. Feasibility of Continuous Monitoring

Of a total of 1920 scheduled hours (h), the total monitoring up-time was 1565 h. The median duration with at least one device recording was 80 h [IQR 50–93] (83% of the theoretical maximum monitoring time) (Table 2); 14 (70%) patients were monitored for more than 2/3 of the planned 96 h.

We observed that nine (45%) patients discontinued using one of or both the SpO_2_ and BP devices during the monitoring period due to discomfort and limited mobility during exercise. Future versions of sensor devices are expected to improve compliance; e.g., a cuffless BP device will be an essential improvement.

### 3.3. Abnormal Vital Signs

All patients had abnormal vital signs at some point during monitoring. The median cumulative total duration of any abnormal vital signs was 620 min [IQR 336–930]. Analyses that were stratified by the actual monitored duration found that the longest cumulative durations were SpO_2_ <92%, with a median of 313 min/24 h [IQR 149–494], and SpO_2_ <88%, with a median of 77 min/24 h [IQR 32.0–123] (Table 3).

The longest median duration of abnormal respiratory rates with RR < 11 min^−1^ was 28 min/24 h [IQR 4.0–121]. The cumulative durations of various circulatory vital signs were reduced compared to those of respiratory vital signs, ranging from medians of 0 min/24 h (all SBP thresholds) to 20 min per 24 h [IQR 3.0–43] for HR > 110 min^−1^ (Table 3). All patients had at least one episode of sustained abnormal vital signs; the most common were desaturation events; 85% had at least one episode of SpO_2_ < 80% ≥ 1 min. Fifty-five percent of the patients had episodes of RR > 30 min^−1^ ≥ 1 min.

The transitional care facility did not measure any vital signs during the monitoring period, but 55% of patients had their vital signs measured within 30 days of study enrollment. The most frequent abnormal vital sign was RR > 24 min^−1^, occurring in 25% of the patients, while RR > 30 min^−1^ and HR ˃ 110 min^−1^ were recorded in two patients (10%) (Table 3).

### 3.4. Complications and Readmissions

Eight patients (40%) developed at least one complication within 30 days of admission to the transitional care facility, including hyperglycemia, infection, and postoperative bleeding.

### 3.5. Patient Case

Figure 3 illustrates vital sign development during a patient’s monitoring period: several desaturation minutes (SpO_2_ < 92% for a total of 1366 min), minutes of tachypnea (RR > 24 min^−1^ for 544 min), and tachycardia episodes (HR > 110 min^−1^ for 2790 min and HR >130 min^−1^ for 818 min). The specific patient, an 83-year-old female, was admitted to the transitional care facility for physical rehabilitation following osteosynthesis of a femoral neck fracture. Eleven days after the continuous monitoring terminated, she was readmitted to the hospital with a urinary tract infection, severe dehydration, pulmonary embolism, and anemia (blood hemoglobin, 5.6 mmol/L). The transitional care facility measured vital signs seven times during the days before the readmission. No interventions were carried out prior to readmission. The length of the readmission stay was 29 days, whereafter the patient was discharged to the transitional care facility again.

## 4. Discussion

The current feasibility study showed that despite a low-staffed environment, CVSM was feasible in a median of 83% of the scheduled time with at least one device recording. The high rate of monitoring was mainly due to the ECG patch (median of 80% recording of the scheduled time), whereas the SpO_2_ device recorded only 33% of the scheduled time. The variation in recording time and missing data is comparable to that reported in other studies using the same devices [13,16,22]. Thus, in an observational method comparison study, continuously measuring HR and RR with the wearable devices SensiumVitals, HealthPatch, EarlySense, and Masimo, found missing data ranging from 13% to 34% for RR and 6% to 27% for HR [23].

One possible explanation for variation differences is the devices’ ease of use. First, the SpO_2_ device did not store recordings locally for later automatic transfer if patients were outside the Bluetooth range, resulting in the deletion of valuable records. Second, the ECG patch adhered to the skin and remained in place for the following 4 days. In comparison, patients reported increased discomfort with the SpO_2_ and BP devices (wrapped around the finger, becoming warm, tightening around the upper arm) and are probably more likely to remove them, as demonstrated by 30% of the patients removing the devices preliminarily. Furthermore, we do not know how the feasibility will be affected by longer durations but expect it to decline.

The median cumulative duration of abnormal vital signs exceeded 191 min per 24 h of monitoring. The prolonged desaturation periods, such as that for SpO_2_ <92%, with a median of 313 min/24 h, were somewhat expected based on previous reports in hospitalized patients [13,16,22,24]. Again, these findings occurred despite the relatively low SpO_2_ monitoring time of 33%.

Based on the above findings, a scalable approach, starting with ECG patches and adding additional devices in the event of deterioration, may be more feasible. This emphasizes the critical nature of comprehending the environment in which patient monitoring occurs.

Although recovery in a transitional care facility may have several advantages (single rooms, improved sleep, and a greater emphasis on physiotherapy), the centers are not designed, equipped, or staffed to detect and treat deteriorating patients who are declining, and the risk of missing late severe complications increases when vital signs are not routinely monitored. The results support this, with few abnormal vital signs being recorded by the transitional care facility in the 30-day period after study inclusion. This was further highlighted in the patient with prolonged periods of respiratory and circulatory abnormal vital signs. Such a patient would have been flagged for diagnostic work-up given the same vital sign deviations.

The complication and readmission rate of 40% is comparable to studies exploring the complication and readmission frequency in patients [4,25,26], considering that most patients had at least two comorbidities. Furthermore, a recent Danish cohort study found that 20% [95% CI 20.1–20.6%] of elderly patients with short hospital admissions were readmitted within 30 days and that heart failure, chronic obstructive pulmonary disease, dehydration, etc., were associated with increased risk of readmission [27]. Severe complications are associated with an increased risk of mortality [3], and our results corroborate a previous study regarding discharge disposition. Elderly patients discharged to an institutional care facility had a higher 30-day (4.3% vs. 0.4%), 90-day (12.6% vs. 1.4%), and 1-year (22.2% vs. 5.9%) postoperative mortality compared to those discharged home with self care [28]. Finally, recent studies have shown the superiority of CVMS not only in detecting deviations but also in significantly impacting patient outcomes regarding reducing adverse events bot in a large case–control study [18] and a recent RCT with 10% absolute adverse event reduction resulting from CVSM vs. intermittent manual monitoring [19].

### 4.1. Strengths and Limitations

This study’s key strength is the continuous monitoring of vital signs in a transitional care facility using previously validated wearable devices [13,20,22], as previous research in this setting is limited. Even so, this study is only a feasibility trial; therefore, our small sample size precludes us from drawing firm statistical conclusions. Further, as this was an observational study, we did not wish for care staff to divert from standard protocol, which could potentially bias results. If the care staff were formally introduced to the devices, encouraged to take ownership of the study, and motivated the patients to keep the devices on, the “up-time” of the SpO_2_/PR and BP devices could possibly have been increased. Another limitation relates to potential selection bias. Of 61 patients screened, 21 declined to participate. As we have no demographic or clinical data on those who declined, we cannot determine whether they were older or frailer or otherwise differed systematically from those enrolled. If non-participants were more impaired, our results may overrepresent the healthier segment of the transitional care population. The direction and extent of this potential selection bias therefore remain uncertain. In a more extensive study, it is a priority to include more patients, ensuring diversity and thus ensuring external validity. However, our included patients all had one or more comorbidity and were combined surgical and medical patients. Despite the limited sample size, this study provides valuable insights into the importance of wear-and-forget devices, and future research should focus on strategies to improve usability and hence participation, such as in extended qualitative studies including patient questionaries.

### 4.2. Future Research

The feasibility of CVSM at the transitional care facility raises hopes for early identification of complications [29] to reduce the severity and high complication rate and the readmission rate to be similar to those in a hospital setting. A key feature will be the potential to remove false alerts, especially in settings where staff resources are scarce. Artificial intelligence-augmented alert generation directly to staff smartphones is part of the ongoing WARD project, where high (>98%) sensitivity is retained whilst removing the majority of alerts [30].

## 5. Conclusions

CVSM was feasible 83% of the potential time at the transitional care facility and identified frequent severe abnormal vital signs, which were undetected by the usual standard of care. The high complication and readmission rate suggest the potential for improvement, which may be aided by CVSM.

## Figures and Tables

**Figure 1 sensors-25-06970-f001:**
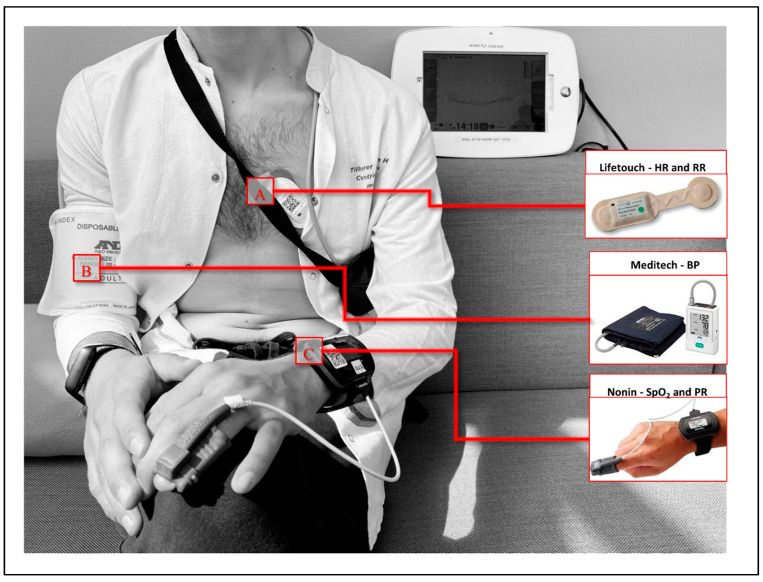
Illustration of monitoring devices.

**Figure 2 sensors-25-06970-f002:**
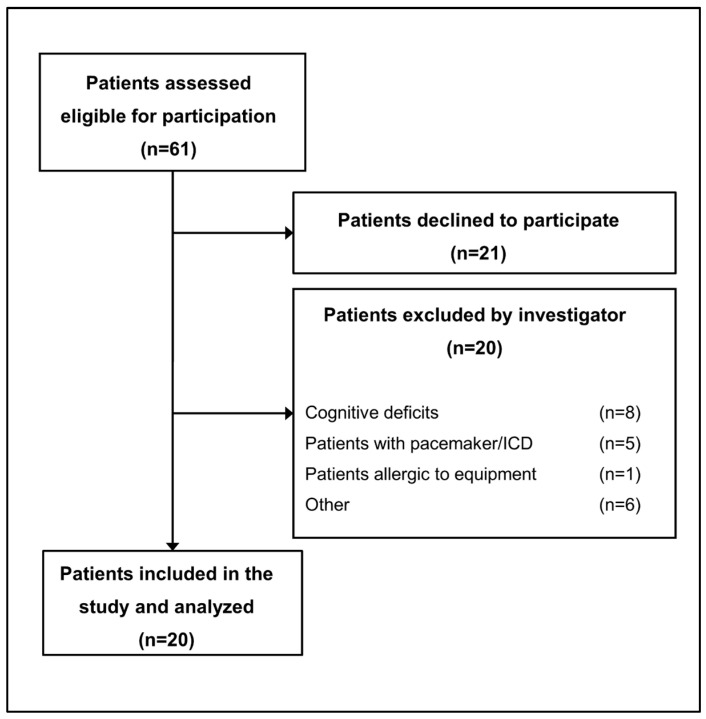
Consort diagram.

**Figure 3 sensors-25-06970-f003:**
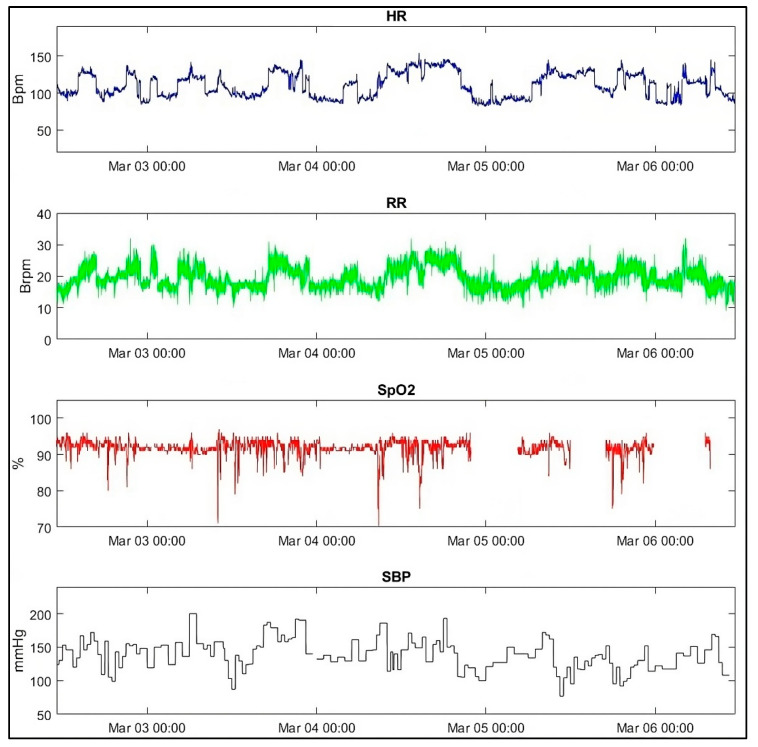
Illustration of a single patient’s vital signs during a monitoring period.

**Table 1 sensors-25-06970-t001:** Demographics.

Parameter	n = 20
**Gender, male, female**	8 (40%), 12 (60%),
**Age, years**	83 [74–89]
**BMI kg/m^2^**	25 [20–29]
**Smoking history (never/previously/current)**	9/9/2
**Excessive alcohol intake**	6 (30%)
**Reason for hospital admission, surgery/medical**	9 (45%)/11 (55%)
**Medical history**	
**One comorbidity**	5 (25%)
**≥2 comorbidity**	15 (75%)
**Stroke or transitory ischemic attack**	4 (20%)
**Epilepsy**	1 (5%)
**Chronic obstructive pulmonary disease**	5 (25%)
**Myocardial infarction**	1 (5%)
**PCI**	1 (5%)
**Atrial fibrillation**	5 (25%)
**Hypertensio arterialis**	5 (25%)
**Congestive heart failure**	4 (20%)
**Hypercholesterolemia**	2 (10%)
**Diabetes mellitus**	3 (15%)
**Chronic kidney disease**	1 (5%)
**Other disease**	13 (65%)
**HR, bpm**	77 [69–86]
**RR, brpm**	16 [14–18]
**SpO_2_, %**	97% [96–98]
**Systolic blood pressure, mmHg**	132 [115–147]
**Diastolic blood pressure, mmHg**	71 [61–78]

BMI: body mass index; excessive alcohol intake is alcohol consumption more than recommended by the Danish Health Authority, which is 24 g/day for men and 12 g/day for women; PCI: percutaneous coronary intervention; mmHg: millimeter mercury. Values are given as numbers (percentage) or medians of [IQR].

**Table 2 sensors-25-06970-t002:** Vital signs measured with one and more devices compared to a theoretical maximum monitoring time of 96 h per patient.

	Median h [IQR]	% of Theoretical Maximum Monitoring Time (96h)
**Any device**	80 [50–93]	83% [52–97%]
**Lifetouch (HR and RR)**	77 [46–92]	80% [48–96%]
**Nonin (SpO_2_ and PR)**	32 [12–63]	33% [13–66%]
**Lifetouch + Nonin (HR, RR, SpO_2_, and PR)**	20 [8–56]	21% [21–58%]
**BP (planned 156 times per 96 h)**	41 times [14–99]	26% [9–63%]

Values are given as numbers in hours, medians [IQR], and percentages (%).

**Table 3 sensors-25-06970-t003:** Duration and frequency of abnormal vital signs over the entire monitoring period.

Monitoring by Continuous Wearable Devices in the Entire Monitoring Period				Manual Monitoring by Transitional Care Facility (n = 11)	
**Minutes of deviations**		**Sustained deviations**			
	Median duration (min/24 h of monitoring)	Sustained deviation	Number of patients (%)	During monitoring period	30 days following inclusion
**Respiratory vital sign abnormalities**					
**SpO_2_ < 92%**	313.0 [149.0–494.0]	**SpO_2_ < 92% ≥ 60 min**	5 (25)	0 (0)	0 (0)
**SpO_2_ < 88%**	77.0 [32.0–123.0]	**SpO_2_ < 88% ≥ 10 min**	9 (45)	0 (0)	1 (5)
**SpO_2_ < 85%**	25.0 [11.0–56.0]	**SpO_2_ < 85% ≥ 5 min**	11 (55)	0 (0)	1 (5)
**SpO_2_ < 80%**	6.0 [2.0–15.0]	**SpO_2_ < 80% ≥ 1 min**	17 (85)	0 (0)	0 (0)
**RR < 5min^−1^**	0.0 [0.0–0.0]	**RR < 5min^−1^ ≥ 1 min**	7 (35)	0 (0)	0 (0)
**RR < 11 min^−1^**	28.0 [4.0–121.0]	**RR < 11 min^−1^ ≥ 5 min**	7 (35)	0 (0)	0 (0)
**RR > 24 min^−1^**	10.0 [2.0–46.0]	**RR > 24min^−1^ ≥ 5 min**	9 (45)	0 (0)	5 (25)
**RR > 30 min^−1^**	0.0 [0.0–2.0]	**RR > 30 min^−1^ ≥ 1 min**	11 (55)	0 (0)	2 (10)
**Circulatory vital sign abnormalities**					
**HR < 30 min^−1^**	0.0 [0.0–0.0]	**HR < 30/min ≥ 5 min**	0 (0)	0 (0)	0 (0)
**HR < 40 min^−1^**	0.0 [0.0–0.0]	**HR < 40/min ≥ 5 min**	0 (0)	0 (0)	0 (0)
**HR > 110 min^−1^**	20.0 [3.0–43.0]	**HR > 110/min ≥ 60 min**	1 (5)	0 (0)	2 (10)
**HR > 130 min^−1^**	0.0 [0.0–3.0]	**HR > 130/min ≥ 30 min**	1 (5)	0 (0)	0 (0)
**SBP < 70 mmHg**	0.0 [0.0–0.0]	**SBP < 70 mmHg ≥ two times**	0 (0)	0 (0)	0 (0)
**SBP < 90 mmHg**	0.0 [0.0–0.0]	**SBP < 90 mmHg ≥ two times**	2 (10)	0 (0)	1 (5)
**SBP > 180 mmHg**	0.0 [0.0–10.0]	**SBP > 180 mmHg ≥ two times**	4 (20)	0 (0)	0 (0)
**SBP > 220 mmHg**	0.0 [0.0–0.0]	**SBP > 220 mmHg ≥ two times**	0 (0)	0 (0)	0 (0)

Values are median [IQR] or numbers (percentage).

## Data Availability

The participants of this study did not give written consent for their data to be shared publicly, so due to the sensitive nature of the research supporting data is not available.

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
