# Peer review of "Continuous Wireless Vital Sign Sensors for Detecting Severe Deviations at a Transitional Care Facility—An Observational Feasibility Study"

_sensors, 2025, doi:10.3390/s25226970_

Round 1
Reviewer 1 Report
Comments and Suggestions for Authors
The absence of hospital-level routine vital sign monitoring in understaffed transitional care facility poses a risk, as it can result in missed patient deterioration and consequently higher rates of severe complications and readmissions. In this manuscript, the authors compared the effectiveness of continuous wearable vital sign monitoring (CVSM) using wearable devices against manual monitoring in detecting abnormal vital signs, while also documenting complication and readmission rate. By employing wearable devices (Isansys Lifetouch, Meditech BlueBP-05, Nonin) to continuously track heart rate, respiratory rate, blood pressure, oxygen saturation and pulse rate in 20 patients for up to 96 hours, the study demonstrates that CVSM is feasible in resource-limited transitional care setting and can identify frequent, severe abnormal vital sign that would otherwise go undetected by standard care. The high rate of complication and readmission further underscore significant shortcomings in current monitoring practice and this manuscript highlights the considerable clinical potential of CVSM to address these problems. The study is well-designed, and with the following minor modifications, I recommend it for publication in the Sensors.
1、In the Abstract section, it is recommended that the authors add a colon (:) after headings Objective, Design, Main outcome measures, Results, and Conclusion. This will prevent potential misinterpretation caused by adjacent words and improve readability. For instance, on line 41, "ObjectiveAfter hospital admissions" should be revised to "Objective: After hospital admissions".
2、In the Introduction section (lines 68-69), the manuscript cites data from 2004 concerning U.S. Medicare patient readmission to contextualize the severity of the problem. As this data is approximately two decades old, it appears somewhat outdated. It is suggested that the authors supplement or replace this citation with more recent literature on readmission rate or associated economic burden in similar patient population to enhance the timeliness and persuasiveness of this study's background.
3、In the Results section, Table 3 appears to suffer from formatting issue, potentially leading to incomplete data presentation or awkward text wrapping. The authors are advised to adjust the table size and layout to ensure all data is displayed completely and clearly.
4、In the Results section (lines 227-229), the text states: "...while RR >30 min-1 and HR ≥110 min-1 were recorded in two patients (10%) (Table 3)." However, the specific data is not explicitly reflected in Table 3. The authors must carefully verify and ensure consistency between the data presented in the text and the table.
Author Response
… The high rate of complication and readmission further underscore significant shortcomings in current monitoring practice and this manuscript highlights the considerable clinical potential of CVSM to address these problems. The study is well-designed, and with the following minor modifications, I recommend it for publication in the Sensors.
Thank you for the kind comments. Also thank you for the work in reviewing our article. We hope that the edits below are satisfactory.
|
In the Abstract section, it is recommended that the authors add a colon (:) after headings Objective, Design, Main outcome measures, Results, and Conclusion. This will prevent potential misinterpretation caused by adjacent words and improve readability. For instance, on line 41, "ObjectiveAfter hospital admissions" should be revised to "Objective: After hospital admissions". |
Apologies for the issues. This was caused by the formatting from one layout to another that was not caught before submission. We have edited it appropriately. |
|
In the Introduction section (lines 68-69), the manuscript cites data from 2004 concerning U.S. Medicare patient readmission to contextualize the severity of the problem. As this data is approximately two decades old, it appears somewhat outdated. It is suggested that the authors supplement or replace this citation with more recent literature on readmission rate or associated economic burden in similar patient population to enhance the timeliness and persuasiveness of this study's background. |
Thank you for the comment. We have updated the numbers with a NEJM article using data from 2015. This article however does not give an estimate of the price of readmissions. For that reason, we have estimated the total price based on a price estimate in 10.3390/healthcare12070750.
Thus the calculation is: (17.8 % of 929.244 + 13.1% of 5.876.773) * $16.037 = $14.998.887.552
Manuscript edit, Introduction lnn. 68-74: In 2015, an analysis of 6.8 million hospital admissions, found hospital readmissions within 30-days was between 13% - 18 %. [1]. This is consistent with previous research demonstrating an up to 16–22 % readmission risk for patients discharged within 30 days of acute or subacute surgery[2–4]. A systematic review using a Monte Carlo simulation estimated the mean price of a readmission to be USD 16,037. That results in an annual cost of approximately USD 15 billion for readmissions alone.[5] |
|
In the Results section, Table 3 appears to suffer from formatting issue, potentially leading to incomplete data presentation or awkward text wrapping. The authors are advised to adjust the table size and layout to ensure all data is displayed completely and clearly. |
We apologize for the formatting issues. Due to a lot of cells, the table was initially fitted on a page in landscape orientation. We have re-formatted the table, to properly fit a standard portrait-oriented page. |
|
In the Results section (lines 227-229), the text states: "...while RR >30 min-1 and HR ≥110 min-1 were recorded in two patients (10%) (Table 3)." However, the specific data is not explicitly reflected in Table 3. The authors must carefully verify and ensure consistency between the data presented in the text and the table. |
We apologize for this data not being readable in the table in the first edition. The table should now be updated to reflect this. We have also corrected a slight spelling error and a sign error for consistency. |
Reviewer 2 Report
Comments and Suggestions for Authors
Comments to Author:
- The manuscript states that the aim is to “evaluate the feasibility of CVSM in transitional care facilities,” but also hypothesizes that it can “detect more abnormal vital signs than standard monitoring.” These reflect two distinct objectives—feasibility and efficacy. However, the study design only supports the former, as no control group was included to test the latter. The title, abstract, and discussion should be aligned accordingly, clearly distinguishing a feasibility study (focused on adherence and device uptime) from a comparative efficacy study (focused on detection performance).
- The sample size (n=20) is small, with no sample size calculation provided. Only descriptive statistics are reported, making it difficult to assess feasibility metrics with confidence intervals or to support the stated hypothesis. Even for a feasibility study, justification for choosing 20 participants should be provided.
- In the introduction, when discussing the importance of continuous wireless vital sign monitoring, the authors may consider citing relevant work on wearable ultrasound systems (e.g., DOI: 10.1109/TUFFC.2024.3492197) to enrich the background context.
- The Methods section should clarify inclusion/exclusion numbers and discuss the potential direction of selection bias (e.g., whether participants who declined were older or had more severe conditions).
- The definition of “monitoring uptime” should be explicitly stated—does it include periods of data loss or transmission interruption?
- Table 3 is incorrectly formatted and appears incomplete. The authors should revise and ensure all relevant data are presented clearly.
Author Response
Thank you for your help in strengthening our manuscript! We have taken extra care in detailing patient inclusion and the sample size decision.
|
The manuscript states that the aim is to “evaluate the feasibility of CVSM in transitional care facilities,” but also hypothesizes that it can “detect more abnormal vital signs than standard monitoring.” These reflect two distinct objectives—feasibility and efficacy. However, the study design only supports the former, as no control group was included to test the latter. The title, abstract, and discussion should be aligned accordingly, clearly distinguishing a feasibility study (focused on adherence and device uptime) from a comparative efficacy study (focused on detection performance). |
Thank you for picking this up. We agree and have changed the abstract, introduction and discussion accordingly with the following edits: Abstract: Removed the fragment ‘compared between’ and added that we only describe data. Changed from: “The transitional care facility staff assessed vital signs in only 11 patients (55 %) without recording any abnormal vital signs.” To: “The transitional care facility staff did not assess vital signs in any patients during the monitored period.” Removed from introduction … and detect a higher frequency of abnormal vital signs than standard monitoring in a transitional care facility. Removed from discussion: The study highlighted the potential need for better monitoring at the transitional care facility based on… |
|
The sample size (n=20) is small, with no sample size calculation provided. Only descriptive statistics are reported, making it difficult to assess feasibility metrics with confidence intervals or to support the stated hypothesis. Even for a feasibility study, justification for choosing 20 participants should be provided. |
We thank the reviewer for this important point. This study was planned and executed as an exploratory feasibility study rather than a hypothesis-testing trial; therefore a formal power calculation for efficacy was not appropriate. We selected a pragmatic sample size of 20 patients to obtain preliminary estimates of feasibility metrics (recruitment rate, device adherence, data completeness, and safety signals) within the available recruitment window and resources. Manuscript edit: Due to the exploratory and hypothesis-generating nature of this feasibility study, not intended to test clinical effectiveness, we did not perform a formal power calculation for an effect size. Instead we selected a pragmatic sample size of 20 participants to obtain preliminary estimates of feasibility outcomes (recruitment and retention rates, device adherence, data completeness, and safety signals) within our available recruitment period and resources. |
|
In the introduction, when discussing the importance of continuous wireless vital sign monitoring, the authors may consider citing relevant work on wearable ultrasound systems (e.g., DOI: 10.1109/TUFFC.2024.3492197) to enrich the background context. |
We thank you very much for the suggestion and have chosen to include a recent review from Sensors on the topic: Bignami EG, Fornaciari A, Fedele S, Madeo M, Panizzi M, Marconi F, Cerdelli E, Bellini V. Wearable Devices in Healthcare Beyond the One-Size-Fits All Paradigm. Sensors (Basel). 2025 Oct 20;25(20):6472. doi: 10.3390/s25206472. PMID: 41157526. |
|
The Methods section should clarify inclusion/exclusion numbers and discuss the potential direction of selection bias (e.g., whether participants who declined were older or had more severe conditions). |
Thank you for this suggestion. We believe the requested information on exclusion/inclusion is already presented in figure 2, but have written this in the beginning of the results section. We do not have information on those who declined, as their information cannot be used. The discussion has also been restructured to accommodate this point.
Results edit: We screened 61 potentially eligible patients, of which 21 patients refused to participate per the ethical guidelines, and 20 patients were excluded by the investigator. The remaining 20 patients were found eligible and consented to participate (Figure 2).
Discussion - Limitations edit Another limitation relates to potential selection bias. Of 61 patients screened, 21 declined participation. As we have no demographic or clinical data on those who declined, we cannot determine whether they were older, frailer, or otherwise differed systematically from those enrolled. If non-participants were more impaired, our results may overrepresent the healthier segment of the transitional care population. The direction and extent of this potential selection bias therefore remain uncertain. |
|
The definition of “monitoring uptime” should be explicitly stated—does it include periods of data loss or transmission interruption?
|
Monitoring up-time does not include periods of data loss or transmission interruption. We have added details accordingly to our manuscript. Manuscript edit: Up-time only included periods with data transmission of valid data. We did not include duration with intermittent data loss. |
|
Table 3 is incorrectly formatted and appears incomplete. The authors should revise and ensure all relevant data are presented clearly. |
We apologize for the formatting issues. Due to a lot of cells, the table was initially fitted on a page in landscape orientation. We have re-formatted the table, to properly fit a standard portrait-oriented page. |